# Critical Review: Propensity of Premise Plumbing Pipe Materials to Enhance or Diminish Growth of *Legionella* and Other Opportunistic Pathogens

**DOI:** 10.3390/pathogens9110957

**Published:** 2020-11-17

**Authors:** Abraham C. Cullom, Rebekah L. Martin, Yang Song, Krista Williams, Amanda Williams, Amy Pruden, Marc A. Edwards

**Affiliations:** 1Civil and Environmental Engineering, Virginia Tech, 1145 Perry St., 418 Durham Hall, Blacksburg, VA 24061, USA; accullom@vt.edu (A.C.C.); martinrl@vmi.edu (R.L.M.); ys117@vt.edu (Y.S.); apruden@vt.edu (A.P.); 2Civil and Environmental Engineering, Virginia Military Institute, Lexington, VA 24450, USA; 3TechLab, 2001 Kraft Drive, Blacksburg, VA 24060, USA; kwilli@vt.edu; 4c/o Marc Edwards, Civil and Environmental Engineering, Virginia Tech, 1145 Perry St., 418 Durham Hall, Blacksburg, VA 24061, USA; k78bass@gmail.com

**Keywords:** non-tuberculous mycobacteria, *Pseudomonas*, *Acinetobacter*, amoebae, copper, iron, PEX, PVC, drinking water, disinfection

## Abstract

Growth of *Legionella pneumophila* and other opportunistic pathogens (OPs) in drinking water premise plumbing poses an increasing public health concern. Premise plumbing is constructed of a variety of materials, creating complex environments that vary chemically, microbiologically, spatially, and temporally in a manner likely to influence survival and growth of OPs. Here we systematically review the literature to critically examine the varied effects of common metallic (copper, iron) and plastic (PVC, cross-linked polyethylene (PEX)) pipe materials on factors influencing OP growth in drinking water, including nutrient availability, disinfectant levels, and the composition of the broader microbiome. Plastic pipes can leach organic carbon, but demonstrate a lower disinfectant demand and fewer water chemistry interactions. Iron pipes may provide OPs with nutrients directly or indirectly, exhibiting a high disinfectant demand and potential to form scales with high surface areas suitable for biofilm colonization. While copper pipes are known for their antimicrobial properties, evidence of their efficacy for OP control is inconsistent. Under some circumstances, copper’s interactions with premise plumbing water chemistry and resident microbes can encourage growth of OPs. Plumbing design, configuration, and operation can be manipulated to control such interactions and health outcomes. Influences of pipe materials on OP physiology should also be considered, including the possibility of influencing virulence and antibiotic resistance. In conclusion, all known pipe materials have a potential to either stimulate or inhibit OP growth, depending on the circumstances. This review delineates some of these circumstances and informs future research and guidance towards effective deployment of pipe materials for control of OPs.

## 1. Introduction

Legionnaires’ Disease is the “leading cause of reportable waterborne illness” in the United States [1,2], with 52,000–70,000 cases per year [1,3,4], 8000–18,000 hospitalizations [5], an overall mortality rate of 15% [4], and high healthcare and legal costs [2,6,7,8]. Bacteria belonging to the genus *Legionella* are the causative agent of Legionnaires’ disease and Pontiac Fever, which infect the human respiratory system via inhalation or aspiration. *Legionella* is classified as “opportunistic” because it preferentially infects those with underlying illnesses or weakened immune systems [4,8,9]. To date more than 60 *Legionella* species have been identified [10], with *Legionella pneumophila* being the species most commonly attributed to human disease [11]. *Legionella* can be found even in “the most aggressively treated drinking water” [12]. Studies have confirmed that potable water is a key source of infection [1,4,13,14,15,16,17], for both hospital- and community-acquired cases [18,19,20]. Other opportunistic pathogens (OPs) such as nontuberculous mycobacteria (NTM), *Pseudomonas aeruginosa*, and *Acanthamoebae*, can similarly be transmitted via tap water and tend to infect individuals belonging to certain risk groups [8].

To infect humans, *Legionella* and other OPs must be present in tap water at the point of use. While *Legionella* can occasionally survive drinking water treatment and be transported through the main water distribution system, the primary environment for *Legionella* proliferation to numbers needed to infect humans generally occurs in building or “premise” plumbing [21,22]. Premise plumbing includes the service pipe that connects buildings to the water main, in addition to the full array of components comprising cold and hot portions of a building’s potable water system [8]. Premise plumbing is characterized by high surface area to volume ratios, longer stagnation times, low disinfectant residual, areas with excess sediment and scale, chemically and biologically reactive plumbing materials, and water with relatively warm temperatures. Such conditions can create ideal micro- and macro-environmental niches for growth of various OPs [1,8,23]. 

Premise plumbing is a key conduit for human exposure via showering, handwashing, and other applications that create airborne aerosols [24]. *Legionella* has been detected in faucets, showerheads, decorative fountains, grocery store mist systems, ice machines, and cooling towers [13,14,16,25]. Larger buildings with more complex plumbing systems are more likely to create physicochemical conditions suited for *Legionella* proliferation, but it is also often detectable in water mains and residences with simple conventional hot and cold water plumbing systems [17,26,27]. A Centre for Disease Control (CDC) summary of Legionnaires’ Disease potable water outbreak investigations from 2000–2014, concluded that 85% of the cases had “deficiencies” in water system maintenance within buildings as a contributing factor [28] and that water chemistry flowing into buildings is one, but not the only, predictor of *Legionella* incidence [29,30]. 

The mechanisms by which premise plumbing influences *L. pneumophila* and other OPs, as well as the broader premise plumbing microbiome, are varied and complex (Figure 1). The influent water chemistry has been found to influence *Legionella,* and also strongly shape the plumbing microbiome, especially through the delivery of growth-promoting nutrients, growth-inhibiting disinfectants, and influent microorganisms [31,32,33,34]. The ecological interactions among microorganisms in biofilms of building plumbing systems can also help overcome barriers to growth from low nutrient levels and disinfectants [24,35,36]. Conversely, other interactions, such as competition, exclusion, predation, or inactivation of symbiotic organisms, may inhibit the growth of OPs [37]. The selective pressures in premise plumbing might also alter the physiologies of resident microbes in a manner that influences infectivity [38]. All these phenomena are further complicated by the fact that premise plumbing configurations, hydraulics, temperature, and water use patterns including velocity, flow or stagnation events, all differ significantly from building to building. In particular, there is strong variability due to occupancy, building size, water heater design, water saving devices, storage and other factors [39,40]. Thus, while there are many overarching similarities, every premise plumbing system is at least as variable as the occupants’ unique water use patterns and habits.

The type of pipe material can also strongly influence the relationship between premise plumbing materials and OPs through both direct effects (interaction with chemical species released from pipe) and indirect effects (secondary consequences of released material from pipes) by altering the level of nutrients, disinfectants, and microbial biomass (Table 1, Figure 1). Selection of pipe material can therefore strongly affect chemistry, biological stability [41], and microbiome composition [42] of the drinking water. 

Motivations for this review include:Growing direct or indirect potable water reuse, which can sometimes alter levels of nutrients and Cu^+2^ in the source water [67].Increased natural organic matter (NOM) in some source waters as an indirect consequence of improving sulfur and nitrogen air pollution controls under rules and regulations such as the U.S. Clean Air Act or Directive 2008/50/EU [68,69,70].Emphasis on and investment in green building design for water and energy efficiency and associated unintended consequences for in-building hydraulics (e.g., more stagnation, higher surface area to volume ratios of water to plumbing surfaces, required hot water recirculation systems) that alter water chemistry and delivery of nutrients or disinfectants [39,54,71,72].Greater use of plastic pipes (e.g., PEX, PVC, polyethylene), which vary in leaching potential by type of plastic and due to the presence of proprietary stabilizers and processes [73].Increasing awareness of viable-but-non-culturable (VBNC) bacteria, which are difficult to measure directly. Molecular and fluorescence-based techniques suggest that they can be prevalent under certain circumstances [8,74] and recent evidence indicates they can still cause disease [75,76].Heightened concern about an array of bacterial OPs besides *Legionella*, including *Pseudomonas aeruginosa, Acinetobacter baumannii,* and NTM, as well as amoebae (e.g. *Acanthamoeba*, *Vermamoeba*), which can themselves be pathogenic or can serve as host organisms for bacterial OP proliferation [8].

Here we critically examine existing knowledge with respect to the direct (Section 2) and indirect (Section 3) effects of common metallic (copper, iron, zinc, aluminum, magnesium) and plastic (PVC, PEX) building pipe materials on the growth of *Legionella* and other OPs, in addition to identifying the complex effects of plumbing system configuration (Section 4) and the characteristics of the drinking water microbiome (Section 5). This review is particularly timely, at a moment when societal expectations for public health protection are elevated and expanding aspirations for improved water/energy conservation will be a major drive of water system design and pipe material selection [39]. In executing this review, we aimed to holistically assess the effects of pipe materials, primarily focusing on *Legionella* while including other OPs, seeking to shed light on why various pipe materials appear to sometimes enhance and other times diminish OP proliferation under real-world premise plumbing conditions. 

## 2. Direct Effects of Plumbing Material on Pathogen Growth

### 2.1. Copper Has Both Antimicrobial and Micronutrient Properties

Copper is sometimes present at trace levels in the source water or in distributed water mains, but the main sources in premise plumbing are copper pipes and brass fittings that are installed beginning at the service line connecting the building to the water main (Figure 2). Due to long-lasting life span, durability, and relatively few concerns about metal release when compared to those of antiquated lead and galvanized iron alternatives, copper and its alloys are common in premise plumbing systems [77]. Copper is a registered antimicrobial of the US Environmental Protection Agency (EPA) [78] and listed as a biocidal product in the European Union, but some countries require special approval for use of copper in drinking water for OP control [79]. It is also an essential nutrient for all living organisms, including humans and OPs [59,80]. Here we review the mechanisms by which copper plumbing may influence control of various OPs (Table 2).

### 2.2. Copper Pipe as an Antimicrobial Material in Premise Plumbing 

The antimicrobial properties of copper were first described more than 3000 years ago in the Hindu Vedas and are occasionally observed at least temporarily in modern plumbing systems [1,120,135,136,137]. The role of supplemental dosing of copper as disinfectants in building plumbing can be important, because *Legionella* and other premise-plumbing-associated OPs are more resistant to chlorine than traditional fecal-associated bacteria that are used for traditional water quality monitoring [8,24,138]. While there is no clear consensus on the primary mechanisms by which copper inactivates bacteria, two hypotheses have been put forward: (1) positively charged Cu^+2^ ions interfere with negatively charged cell membranes, creating holes; and (2) Cu^+2^ disrupts the replication and production of DNA, RNA, and proteins, potentially through metabolic cycling between Cu^1+^ and Cu^2+^ oxidation states, which generates radical oxidative species such as hydroxide radicals [139]. In potable water, copper passively released from plumbing materials can be present in the germicidal range for *Legionella* of 0.1–0.8 mg/L [62,119,120,140], even in some parts of plastic pipe systems connected with brass fittings [141,142]. Passive release or purposeful dosing that results in copper concentrations of 0.05–0.8 mg/L are thought to limit *Legionella* growth [62,83,119,120,143]. 

A number of studies have confirmed the efficacy of copper, either passively leached from premise plumbing materials [59,140,144] or actively added using copper-silver ionization (CSI) systems [62,83,145], as a *Legionella* antimicrobial. Biofilms grown at room temperature for 30 days in pre-sterilized reactors with copper, PVC, and stainless steel coupons were found to have lower total bacterial counts on copper than PVC surfaces [146]. Other batch reactor studies indicate similar results, demonstrating lower *L. pneumophila* numbers on copper plumbing than plastic plumbing [59,140,144,147]. Analogous responses to copper surfaces by other Ops, such as *Klebsiella* spp. [148], NTM [111,149], *P. aeruginosa* [128], and *Aeromonas hydrophila* [114], have been reported. Two different field studies found that copper concentrations were significantly lower in samples positive for *L. pneumophila* than samples negative for *L. pneumophila* [150,151]. Borella et al. [23,152] identified a threshold total copper level of 0.5 mg/L in one sample of water, above which samples were approximately two to seven times less likely to be positive for *L. pneumophila*.

Studies of CSI applications also demonstrate that copper can have direct antimicrobial effects. Lin et al. [83,109] showed that 0.5 and 48 h of exposure to 0.4/0.04 mg/L copper/silver achieved 99% inactivation of *L. pneumophila* and *Mycobacterium avium*, respectively, in bench-scale testing. Stout et al. [119] performed long-term monitoring of CSI systems in 16 hospitals and demonstrated their efficacy for *Legionella* control, as the numbers of hospitals with >30% *Legionella* positive samples dropped from 7/16 to 0/16, and no Legionnaire’s disease cases were reported in 15 out of 16 hospitals after the implementation of CSI. Addition of copper ions to solution from pipes or via CSI, at the bench and building-scale, has also been shown to inhibit the growth or reduce the frequency of OPs such as *Staphylococcus* spp.[98,99], *Stenotrophomonas maltophilia* [91,92,104], *Acinetobacter baumannii* [58,91,92], NTM [108,109], and *P. aeruginosa* [91,92,98,99,127,130]. 

#### 2.2.1. Noteworthy Limitations to Copper’s Antimicrobial Efficacy

Despite the encouraging examples presented in the previous section, the overall success of copper as a disinfectant for *Legionella* is mixed [110]. Several studies have found that the antimicrobial effects of copper were limited, or that copper even encouraged growth of *Legionella* in some instances [63,83,122,153]. In one study, *Legionella* was consistently detected in a hospital hot water plumbing system with average pH = 7.7, even when copper was present at concentrations of 1.1 ± 0.2 mg/L [153]. Other studies have shown similar trends. For instance, Giao et al. [121] found no significant difference between biofilm formed on plastic (PEX and PVC) coupons and biofilms formed on copper coupons when the biofilms contained a heterogeneous community or when the biofilms were purely *L. pneumophila*. *P. aeruginosa* has been found to persist in hospital copper plumbing [129] and the implementation of a CSI system in one hospital did not appear to fully eliminate patient *P. aeruginosa* infections associated with exposures from faucets [130]. 

Prominently, in one field study conducted in Germany with low or no chlorine residual, hot water systems containing copper pipes were colonized with *Legionella* much more often (>30x) than those with galvanized steel or plastic pipes, despite the fact that the temperature of the hot water in these systems was similar. Also, samples (n = 44) from hot water recirculation lines with >0.5 mg/L of copper displayed 2,4000 ± 15,000 (mean ± standard deviation) CFU *Legionella*/L, while samples (n = 153) with ≤0.5 mg/L of copper had 10 ± 100 CFU *Legionella*/L [63]. 

There are many possible explanations for the apparent contradictions in overall impacts of copper (Table 2). It is important to first recognize that the antimicrobial properties of copper can be almost completely controlled by water chemistry (Figure 3). Notably, the concentration of Cu^+2^ and its associated inorganic ions tend to decrease in concentration in aged pipes, at higher pH, or in the presence of common corrosion inhibitors, such as orthophosphate. Unfortunately, studies frequently do not collect or report such relevant data [63,129,130,153], limiting the ability to trace differences in copper’s antimicrobial efficacy to water quality parameters. There is also the likelihood of strain-to-strain differences in copper resistance, and the selection for copper resistant organisms in systems with copper pipes [154,155]. 

#### 2.2.2. Water Chemistry Effects on Copper Bioavailability

The chemistry of the influent bulk water can reduce toxicity of copper by: (1) reducing overall solubility and the equilibrium level of Cu^+2^ in the presence of copper rusts [156,157]; (2) forming copper complexes [158,159,160], (3) having elevated divalent (Ca^2+^, Mg^2+^) or trivalent (Fe^3+^, Al^3+^) cations, which compete with copper for uptake sites of organisms [161,162,163]. Therefore, water chemistry details are useful to explain the discrepancy of copper effects, but such information is often lacking in some studies [63,121,129,130,153].

Prior culture-based research demonstrated that precipitation of copper at pH 9 reduced toxicity of copper towards nascent *L. pneumophila* colonies by 16-fold relative to pH 7, where copper is more soluble [83]. Other compounds known to reduce levels of Cu^+2^ by complexation and precipitation are logically expected to interfere with copper antimicrobial properties and include NOM and either ortho- or poly-phosphates [156,157,158,159,160]. Specifically, NOM and polyphosphate sequestrants can vary in concentration and complexation ability from water to water, can bind Cu^+2^ and dramatically reduce its bioavailability. Orthophosphate added as a corrosion inhibitor can reduce metal pipe corrosion rates and lower free metal ion concentrations in drinking water. For example, our research has shown that the addition of 3 mg/L of phosphate and 5 mg/L NOM at pH = 7 reduced copper’s antimicrobial effects towards *L. pneumophila* by four and seven times, respectively [164]. 

Copper’s antimicrobial properties are expected to increase at lower pH, lower hardness, lower Al^+3^ and Fe^+3^, lower phosphate or polyphosphate, lower NOM, and colder temperatures due to known interactions with Cu^+2^ ion. Studies of copper toxicity to algae and higher aquatic organisms have shown that Mg^2+^, Ca^2+^_,_ Al^+3^_,_ and Fe^+3^ compete with copper for binding sites, reducing the toxicity of copper [161,162,163]. For instance, Ebrahimpour et al. [161] reported that the 96-h median lethal concentration (LC50) values for *Capoeta fusca* increased roughly linearly (1.1 to 7.5 mg/L copper) over a hardness range of 40-380 mg/L as CaCO_3_. Trivalent metal ions, such as Al^3+^ and Fe^3+^, can also form a layer of metal hydroxide gel around cells that can sorb copper and reduce its availability [165]. Free copper also tends to decrease at higher temperature and as pipe scales age [54,166]. 

#### 2.2.3. Copper as a Nutrient in Premise Plumbing

Copper (Cu) is an essential micronutrient used in protein synthesis, respiration, various oxidation/reduction reactions and other functions in prokaryotes [80,167]. Accordingly, it is reasonable to suspect that copper piping might sometimes act as a source of this essential nutrient in premise plumbing, thereby increasing microbial growth relative to other materials. Buse et al. [122] showed that effluent from CDC biofilm reactors equipped with coupons of different pipe materials at pH > 8 and PO_4_ > 0.2 mg/L, had up to 20× more *L. pneumophila* gene copies when copper coupons were used relative to PVC coupons. Mullis et al. [111] indicated that copper surfaces supported two to four times more *Mycobacterium abscessus* than PVC. Mathys et al. [63] reported that hot water systems containing copper pipes were colonized significantly more often than those with galvanized steel or plastic pipes. 

### 2.3. Direct Release of Organic Carbon by Plastics 

Potable water is oligotrophic, because organic carbon is relatively scarce and often limiting to the growth of drinking water microorganisms [24,168,169]. Plastic premise plumbing pipes, which are made with polymeric organic compounds, including stabilizers, flexibilizers and plasticizers, can leach organic carbon to water [56,57,170] whereas metallic pipes do not. These organic carbon compounds can fuel the growth of *Legionella* [45,59] and presumably other OPs. In some cases, the organics leached to water are not the polymers themselves, but rather are additives (i.e., flexibilizers, plasticizers, stabilizers) to improve aspects of pipe performance [42,170,171]. 

New PEX pipes commonly leach 100-1800 µg/L of total organic carbon (TOC) as determined by temperature, stagnation, surface area to volume ratio, pipe brand and age [56,170,172]. These levels of carbon, are far above the commonly cited threshold of 100 µg/L suggested to spur microbial growth in potable water main distribution systems [173]. However, the proportion of this released organic carbon that is assimilable is not clear. Many studies have demonstrated that some PEX pipes increase biofilm growth [59,140,147] and OP growth [59,140] relative to copper and iron. Unfortunately, it is unclear how general these effects are because the formulation of PEX used (e.g., PEX-b) varies from one manufacturer to another [170,172] and is typically proprietary and thus not cited in the available literature [59,140,147]. An experiment in the Netherlands using small-scale recirculating water heater systems (eight gallon tanks) connected to copper or PEX pipes (19.4 ft) attributed over three times higher *Legionella* bulk water levels in PEX pipe systems as compared to copper pipe systems although the authors did not determine if the difference was due to copper antimicrobial effects or leached organic carbon growth-promotion [140]. 

PVC pipes can leach 60–50,000 µg/L of TOC under typical water use conditions [50,56,174], of which roughly 50% was estimated to be assimilable [42]. Other studies indicate that PVC can promote biofilm growth [175,176] and proliferation of OPs compared to copper, lined cement, iron, and stainless steel [111,177,178,179]. When copper, glass, PEX, and PVC were used as materials in a biofilm apparatus simulating premise plumbing, PVC and PEX materials maintained the highest *Legionella* growth potential in remineralized reverse osmosis water [178]. Other studies have drawn similar conclusions for other OPs compared to copper [111,128,148,149].

### 2.4. Iron Release from Pipes

Iron pipes may provide important niches and nutrients for OP growth. Antiquated cast iron, galvanized iron, and steel pipes in service lines and home plumbing can leach iron to water in a range of 0.2–18 mg/L dependent on factors including water chemistry, stagnation, surface area to volume ratio, and historical corrosion control [180,181]. Iron can also accumulate in loose deposit or biofilms and some studies have suggested that such locations are hotspots for growth of *Legionella* and other pathogens [40,182]. Studies examining *M. avium* have found that galvanized steel supported more growth than copper, PVC, and stainless steel [111,149]. 

Iron is an important nutrient for microorganisms involved in oxygen transfer, protein synthesis, and other essential metabolism [183] and some studies have shown that the presence of iron contributes to OP growth. Bench-scale studies have demonstrated that iron concentrations of up to 1 mg/L could enhance *L. pneumophila* growth in tap water while high concentrations (10, 100 mg/L) of iron produced toxic effects on *L. pneumophila* [184]. During the Legionnaires’ Disease outbreak in Flint, MI, our research found that the median iron concentration was 0.11 mg/L in cold water samples during the outbreak, but the outbreak’s end coincided with a water switch, dropping median iron in cold water samples down to less than 0.01 mg/L [26]. Other field studies have observed similar positive correlations between *L. pneumophila* levels and iron concentrations [15,185]. In a simulated household drinking water system with no chlorine, van der Lugt et al. [186] observed that colonization of stainless steel faucets by *Legionella* was enhanced in the presence of 0.09 mg/L cast iron rust. It is important to note that in any study employing chlorine, iron pipe corrosion will remove the chlorine, confounding simplistic attribution of the higher *Legionella* to either iron or chlorine [26,187,188]. One study specifically examined if iron addition increased *L. pneumophila* growth without any chlorine present, and showed that it did so in one water with naturally low iron, but had no effect in another water with relatively high ambient iron [187].

### 2.5. Zinc, Aluminum, Magnesium Plumbing Materials

Pipes and plumbing devices can be composed of other metals that might affect the growth of OPs, but their impacts are largely unexplored. Zinc is present in source waters in concentrations ranging from <0.011 to 0.04 mg/L [189,190] and is normally below 0.1 mg/L in finished water [191]. Zinc concentrations at the tap are largely driven by its addition in corrosion inhibitors, or release from brass fixtures and galvanized pipes [190,191,192], and concentrations can reach 5 mg/L or higher [193,194]. Analogous to copper, zinc is an essential nutrient for microbial growth [195,196,197,198,199,200]. Zinc addition has been shown to increase *L. pneumophila* and *P. aeruginosa* growth in culture media [201], and high soluble zinc has been correlated with NTM [202].

Zinc can be toxic to microorganisms [196,203,204,205,206], but is believed to have limited biocidal activity compared to other metals [207], especially as it is below the US EPA Secondary Drinking Water Regulation limit of 5 mg/L [207] and Chinese Standard for Drinking Water Quality of 1 mg/L [208]. Inhibitory concentrations of zinc for Ops such as *Pseudomonas* spp., *P*. *aeruginosa*, and *Aspergillus niger* range from 13 to 650 mg/L in nutrient broth [204,205,206]. While this is a relatively high concentration range, Zhang et al. [180] demonstrated that galvanized iron pipes can release zinc to these levels in the presence of nitrifying bacteria. Furthermore, the biocidal activity of zinc or any other trace metal in premise plumbing will be controlled by the same chemistry factors including pH, hardness and NOM mentioned previously for copper. 

Aluminum or magnesium rods are also commonly present as sacrificial anodes in water heaters (Figure 4), elevating Al^+3^ or Mg^+2^ levels in the water. Mg^+2^ is known to be an essential nutrient for *Legionella* [201], whereas no such criteria have been established for Al^+3^. More research is needed to determine whether these additional trace metals encourage or discourage OP growth in plumbing systems. 

## 3. Indirect Effects of Pipe Material on Pathogen Growth

### 3.1. Pipe Material Effect on Disinfectant Availability

Pipe material is a key factor affecting disinfectant decay in potable water systems. Maintaining relatively high levels of disinfectant residual is important to OP control because OPs are 20–600x more disinfectant resistant than the common indicator microorganisms such as *E. coli* [24] and are further protected in biofilms or host organisms [209,210,211,212,213,214]. Plastic pipe materials are generally non-reactive with chlorine and chloramine in terms of maintaining disinfectant residual levels, even though chlorine does sometimes slowly react with and degrade certain types of PEX and polyethylene pipe [44,45,46,47,48,49,51,215]. On the other hand, iron pipes have an extremely high disinfectant demand, as free chlorine cannot co-exist in equilibrium with ferrous or zero valent iron [44,46,47,48]. While chloramine is relatively non-reactive, iron oxide scale and associated nitrifying biofilms can cause relatively rapid monochloramine decay [216,217]. The reactivity of copper pipes and copper oxides is typically between plastics and iron and chemically catalyzes both chlorine and chloramine degradation [43,54,156,218,219,220]. Higher pH and the existence of phosphate can help maintain disinfectant residual levels in both iron and copper pipes [26,54].

### 3.2. Effect of Metallic Plumbing Materials on Nutrient Availability via Autotrophic Carbon Fixation 

Although metallic plumbing does not leach assimilable organic carbon directly to water, certain metals can indirectly help OPs overcome carbon limitations by facilitating the growth of autotrophic microorganisms. Specifically, metallic pipes can encourage growth of hydrogen-oxidizing, ammonia-oxidizing, and ferrous-oxidizing autotrophic bacteria that fix inorganic carbon into new biomass [66,221].

#### 3.2.1. Hydrogen Oxidizing Bacteria

The corrosion of iron pipes and the galvanic corrosion of aluminum or magnesium sacrificial anodes protecting steel water heaters can evolve hydrogen gas, which is a strong electron donor for autotrophs [60,61,110,221]. Ishizaki et al. [222] indicated that hydrogen-oxidizing bacteria, *Alcaligenes eutrophus*, could fix 2300 µg C/mmol H_2_ in biomass in closed circuit cultivation system at gas pressure slightly higher than atmosphere, which could practically translate into production of up to 80 µg/L organic carbon biomass per day in an 80-gallon water heater equipped with a magnesium anode [223]. A study by Dai et al. [224] of an experimental water heater plumbing rig at 39, 42, and 51 °C confirmed elevated levels of functional genes associated with hydrogen metabolism, demonstrating that hydrogen-oxidizing bacteria were able to proliferate in water heaters. 

#### 3.2.2. Autotrophic Ammonia and Iron Oxidizing Bacteria

Iron and copper can catalyze the conversion of chloramine disinfectant to free ammonia, which can then serve as a substrate for autotrophic ammonia oxidizing bacteria. Ammonia-oxidizing bacteria can fix substantial amounts of organic carbon into the system, specifically 21 to 240 µg C/mg NH_3_-N based on experimental growth yield values of pure or mixed cultures [225]. Ferrous iron, released as a natural by-product of iron corrosion, can also fix an average of 26 µg C/mg Fe^2+^ under circumneutral condition measured in bioreactors [226]. 

#### 3.2.3. Copper Deposition Corrosion Accelerating H_2_ Evolution

Although copper cannot corrode with evolution of H_2_ gas, cupric ions in water can plate onto the less noble metals (zinc, aluminum, iron and magnesium) via deposition corrosion. This copper coating can dramatically accelerate corrosion of less noble metals and indirectly stimulate evolution of hydrogen (H_2_) gas (Figure 4) [66,222,227,228]. A study using a combination of bench- and pilot-scale hot water system experiments demonstrated these effects [222]. 

### 3.3. Pipe Scaling Effects

Scaling caused by pipe corrosion or higher pH can increase pipe surface roughness, which is known to enhance biofilm colonization and overall growth, creating an ideal environment for OP establishment and proliferation [112]. One study showed that copper coupons in a biofilm reactor formed extensive scales and promoted seven-fold more biofilm biomass than PVC pipes after three months of incubation [230]. Aged metal pipes may form very thick scales characterized by corrosion tubercles and extensive networks of pores [60,231,232,233], providing an area for not only additional biofilm growth, but also distinct microenvironments [233,234] with pH is as low as 2.0 or as high as 10 [235].

## 4. Influence of Plumbing System Design, Configuration and Operation

All of the direct and indirect interactions described in previous sections are further influenced by the specific premise plumbing design, configuration, and operation. Flow rate, water stagnation, temperature profile, secondary disinfectant concentration, and nutrient availability can all interact to create hot spots for OPs growth in buildings. 

### 4.1. Water Stagnation

Water age is defined as the time it takes water to move from one point to another in the system, which may influence OP growth through a variety of mechanisms. This includes the time from when it is freshly produced at the treatment plant and travels to the service line, as well as the time from when it first enters the building’s plumbing to the point of use [71]. High water age in buildings is increased by: (1) existence of dead ends/legs and stagnation in plumbing systems [182,236]; (2) use of low flow devices or presence of large storage tanks such as those used for solar water heating or onsite rainwater collection [39]; and (3) using low volumes of water in a building or at a particular outlet [192]. Stagnation and infrequent water use may concentrate and enhance release of organic matter in water in plastic pipes and metals in metallic pipes [181,237,238,239,240]. Zhang et al. [241] found a four-fold increase in bulk water TOC in unplasticized PVC pipes between 24 h and 72 h of stagnation. Fixtures in a green building with the fewest water use events (most stagnation) also had greater organic carbon, bacteria counts, and heavy metal (Zn, Fe, Pb) concentrations [192,242].

Stagnation and high water age also increases the likelihood and rate of disinfectant decay. High consumption of chlorine and chloramine during stagnant periods of 24–72 h have been observed for synthetic pipes (0.4 and 0.6 mg/L of chlorine loss, respectively), and stagnant periods of 2–8 h in metallic pipes (3 and 4 mg/L chlorine loss, 1.5 and 3.5 mg/L chloramine loss, respectively) [54,241]. In a green building study, six-hour stagnation almost fully eliminated monochloramine (>99%) within pipes [71].

Such water quality changes have been related to increased levels of OPs in premise plumbing systems [39,243,244,245]. In a field sampling study of main water distribution system, 120 water samples were taken throughout a drinking water distribution system. Only four samples were positive for cultivable *L. pneumophila* and all four samples were taken from dead end points at the end of streets with no chlorine residual remaining [246]. Another field study identified their most frequently *Legionella* positive sites as being located at the end of the distribution system and having the highest turbidity, iron, TOC, and water age, as well as the lowest flow [247]. The association between OPs and stagnation has created interest in strategies to reduce building water stagnation effects such as removing dead-legs, flushing, maintaining the hot water system, and shock disinfection [248,249,250,251]. The effectiveness of these strategies should be evaluated within the context of the specific pipe materials that are present.

### 4.2. Hot Water Recirculation Lines 

Some plumbing codes require or suggest the use of recirculating hot water lines for water/energy conservation, convenience and comfort [1,252,253,254]. In these systems, water is circulated continuously between the water heater and the point of use, preventing cooling of the distal lines and allowing for nearly instant delivery of hot water at the point of use [255]. There are many important differences between hot water recirculating systems and conventional systems, which are stagnant during periods of disuse that can affect OP growth. The constantly flowing water can deliver more nutrients to biofilm and hypothetically increase OP growth [230]. On the other hand, continuous flow can deliver more disinfectants and more hot water, which are critical control measures for OPs [256,257]. The net effect depends on which of these factors is dominant. 

Continuously recirculating water could also increase release of metals, increase deposition corrosion of anodes by constantly recirculating water through copper pipe, and result in greater accumulation of sediments and H_2_ gas. One study showed that recirculating systems with copper piping had 3–13 times more aluminum and copper, 4–6 times more hydrogen in effluent water, and 9% more aluminum anode weight loss, compared with standard (non-recirculating) systems [222]. Recirculating systems can also accumulate 3–20 times more sediments [222] arising from corrosion of metallic pipe material and the anode rods [157,232,233,234,258]. These sediments, which also collect at the bottom of hot water tanks, may serve as an important growth niche within warm regions of hot water tanks where influent cold water depresses temperatures, and there are also relatively low levels of disinfectant and high levels of nutrients for *Legionella*, heterotrophs, and host organisms [17,259]. 

### 4.3. Pipe Aging

New plastic and copper pipes behave differently than older pipes. Specifically, corrosion and release of metals is strongly influenced by pipe age, with corrosion rates and metal release tending to decrease as thicker and more passivating pipe scales form. Aging can dramatically reduce levels of metal leaching from copper and other pipes [157,260,261]. The rate of aging, and whether it decreases release of pipe constituents at all, is highly affected by water chemistry and water use patterns [157]. Likewise, leaching of organics from plastic pipe may attenuate 50% to >99% after aging for a period of a few weeks with hot water exposure [51,170], but in other cases has been sustained for months [262] or even over a year [263]. Pipe aging is an important factor to consider when comparing PEX to copper’s capacity for *Legionella* growth. One study showed that the *Legionella* numbers in bulk water of both PEX and copper pipes in a simulated warm water system were the same after two years [140]. We speculate that one possible cause for this convergence is that, as plastic pipes age, organic carbon migration to water decreases, whereas levels of antimicrobial copper released from copper pipe also tends to decrease. Hence, in some situations, it is expected that in very old copper and plastic pipe systems there would be little difference between these pipe materials.

### 4.4. Possible Mixed Material Interactions

Building plumbing is typically comprised of multiple pipe materials, either in the original design or after partial retrofits or renovations. It is anticipated that there are sometimes synergistic and other times antagonistic interactions between pipe materials that would influence growth of OPs. Copper deposition accelerating the evolution of H_2_ from aluminum, zinc, magnesium and iron corrosion, as discussed in Section 3.2.3, is an important exemplar. Copper is also known to catalyze degradation of plastic pipes [264,265,266,267,268], and the presence of copper pipe upstream of plastic pipe might enhance organic carbon release [268], surface roughness for biofilm growth [264], and perhaps even disinfectant consumption due to copper in the scale. Iron pipes upstream of copper may produce mixed Fe-Cu oxides, which can be extraordinary catalysts for free chlorine decay [269]. Similarly, copper released upstream of iron pipes could increase iron release [270]. Any galvanic coupling between two metals in plumbing materials (copper/brass-lead [271,272], copper/iron [270,273,274] iron/zinc [275,276], copper/aluminum [277,278], copper/zinc [271,279], copper/magnesium [280]) has the potential to enhance corrosion and cause changes to water quality parameters relevant to corrosion and OP growth [235,281], dissolved oxygen (DO) [273], metal concentrations [271,272], and disinfectant residual concentration. These reactions also create microenvironments of very high or very low pH [235,238]. Given that in the 2017 American Housing Survey 10% of households that reported any home improvement projects also reported adding or replacing an interior water pipe [282], understanding the effects of mixing pipe materials during renovation appears to be a valuable research area as antiquated premise plumbing is increasingly replaced.

## 5. Mediating Role of Microbiome and other Microbiological Considerations

### 5.1. The Role of Pipe Material in Shaping the Premise Plumbing Microbiome and Resident Amoeba Host Organisms

Interactions between OPs and the microbial communities surrounding them are key to OP proliferation and are likely influenced by pipe materials. OPs can be parasitic to free-living amoebae that first prey upon them in drinking water biofilms, before they reproduce inside and eventually kill the host organism [24]. In fact, there is some doubt that *Legionella* actually reproduces significantly in drinking water outside of an amoeba host [283]. Amoebae can also protect OPs from disinfectants and provide access to nutrients. For example, *Legionella* exclusively use amino acids, which are abundant in amoeba vacuoles, as a carbon source [210,211,212,213,214,284,285]. Thus, although poorly studied, any factor altering growth of key host amoebae (including *Acanthamoeba*, *Vermamoeba*, and *Naegleria)* is expected to indirectly affect growth of OPs, including *L. pneumophila, P. aeruginosa,* and NTM [122,210,211,212,213,214,225,257,286,287]. In one experiment, copper coupons were found to host more *Acanthamoeba polyphaga* than PVC coupons [288], possibly because copper hosts less diverse eukaryotic communities [64,289] and limits competition for *A. polyphaga*. As a result, *L. pneumophila* grew and shed to the bulk waters in higher numbers on these copper coupons than on PVC coupons if co-inoculated with *A. polyphaga* [122].

Interbacterial interactions may also influence the growth of OPs. Broadly speaking, OPs benefit from the biofilm community through access to nutrients and protection from disinfectants [24,35,36,290]. Some studies have identified correlations between specific taxa and OPs in premise plumbing [291], cooling towers [292] and drinking water distribution systems [293]. However, the significance of these correlations to premise plumbing material selection is not well understood, as most studies examining differences in bacterial communities focus on very broad measures of community structure [48,59,64,216,289,294,295,296]. Certain waterborne bacteria are known to produce toxins that inhibit *L. pneumophila* growth [216,297] or exude other compounds that have secondary bacteriostatic effects on *Legionella* [298]. Intra-bacterial inhibition also may be mediated through amoebae by reducing host uptake [299,300] or killing the host population [134,301,302]. More research is needed to elucidate how the broad ecological differences resulting from pipe material influence these interactions. Integration of metagenomic or meta-transcriptomic analyses targeting the production of bacteriocins or other toxins with known effects on OPs could elucidate the ecological effects of taxonomic shifts resulting from pipe material. Interrupting OP-amoeba endosymbiosis through the enrichment of preferential non-OP amoeba prey [299,300] has been suggested as a probiotic means of controlling OPs [303], and pipe material could be explored as a means of enrichment of these taxa.

### 5.2. Variation in Copper Tolerance Among Species and Strains

Strain-to-strain differences in intrinsic tolerance of copper, acclimation to copper concentrations with time through induction of the appropriate genes, or acquisition of copper resistance via mutation or horizontal gene transfer in premise plumbing might explain some of the discrepancies in variable outcomes of copper on OPs (Table 2). *Legionella* [155] and other OPs [58] may acclimate to high copper levels through the expression of copper detoxification or efflux systems. Bedard et al. [155] reported four-fold differences in the copper tolerance of environmentally-isolated *L. pneumophila* strains, noting that more resistant strains showed increased copper ATPase *copA* expression, speculating that their increased tolerance may also be a result of higher biofilm production. Strikingly, Williams et al. [58] showed that, during exposure to 95 mg/L of copper over 6 h in liquid culture, culturable *A. baumannii* levels (CFU/mL) could increase by 2-logs or decrease by 2-logs, depending on the strain. The authors identified putative copper detoxification and efflux systems within the genome of the most resistant isolate and identified specific genes that were upregulated in response to copper exposure. However, a majority of the less tolerant strains tested also possessed these genes, leading the authors to suggest that further definition of the proteins involved in copper resistance is required. One recent study showed two environmentally-isolated *Legionella* strains reduced by less than one log in culturability, even after two weeks of exposure to 5 mg/L copper, which the authors attributed to adaptation to the high levels of copper (average 0.48 mg/L ) in the hot water system from which these isolates were collected [154]. A profile of *Fusarium* isolates revealed that tap water isolates were more copper-tolerant than soil isolates [303]. *P. aeruginosa* isolates isolated from a hospital with copper plumbing exhibited only slightly limited growth in the presence of 0.15 mg/L copper [129]. All of these strains were found to harbor GI-7, a mobile genetic element that confers copper resistance and that has also been identified in a *P. aeruginosa* strain associated with hospital outbreaks [304]. Limited data suggest that *A. baumannii* and mycobacteria are more difficult to inactivate with copper than other OPs, while *P. aeruginosa* is more readily inactivated [91,92,98,108,109]. *L. pneumophila* has been found both at the more resistant [98] and less resistant [91,108,109] ends of this spectrum. The wide variability among OPs and even strains of OPs in their intrinsic tolerance of copper, ability to acquire genetic resistance, and ability to acclimate to elevated levels of copper makes it difficult to precisely predict the efficacy of copper and other antimicrobials for OP control. 

### 5.3. Confounding Effects of VBNC Bacteria 

The discovery of VBNC bacteria has complicated prior understanding for all OP control strategies, including copper. Virtually all prior work relied on culture methods to determine copper’s efficacy for killing OPs [62,63,83,91,92,98,108,109,120,137,153], but some microbes rendered not culturable might remain viable and still infect host amoebae or humans [74,76,305,306,307]. The existence of VBNC pathogens in premise plumbing has been demonstrated by comparing culture-based numbers with those enumerated via fluorescence (e.g., live/dead) and molecular-based (e.g., quantitative polymerase chain reaction) monitoring methods [308]. 

Bench-scale studies examining copper’s antimicrobial efficacy have found discrepancies between culture-based and molecular-based numbers of *L. pneumophila* [121,122] that are also suggestive of a copper-induced VBNC state. Similar discrepancies have been noted for *P. aeruginosa*, *Stenotrophomonas maltophilia,* and *M. avium* [104,109,127,132,133]. Evidence of copper-induced VBNC activity is particularly strong in the case of *P. aeruginosa*, where one study applied multiple non-culture-based measures of viability [127,132]. Furthermore, VBNC *P. aeruginosa* have been shown to partially recover infectivity after removal of copper from solution [132,133]. To understand how VBNC bacteria contribute to OP infections, additional studies are needed to delineate the premise plumbing conditions more precisely that induce VBNC status and to confirm the range of functionality maintained in this state. A primary challenge in achieving this is that there are currently no reliable methods for confidently enumerating VBNC bacteria.

### 5.4. Virulence

The premise plumbing environment exhibits several features that could possibly contribute to the virulence of resident OPs. Wargo [38] describes features of drinking water plumbing that could prime OPs to infect cystic fibrosis patients, although the interactions described in this review could also pose risk to otherwise immunocompromised individuals. Such features that are relevant to pipe material include [38]:Elevated copper levels, selecting for resistance to copper overload within macrophage phagosomes, a component of the innate immune response [309].Elevated iron levels, influencing interactions between iron homeostasis and virulence.Exposure to lipids, which are generally not well removed by drinking water treatment, priming OPs for lipid-rich environments within hosts. Accumulation of phospholipid fatty acids has been shown to be greater in the biofilms of polyethylene pipes than copper pipes, though these lipids were putatively associated with bacteria [310].Low DO levels, selecting for OPs capable of survival in low DO regions of the biofilm in infected host tissue.Exposure to eukaryotic predation, selecting for resistance to the host’s immune response (e.g., lung macrophages) or enhanced virulence.

Some studies suggest that the above types of interaction may increase the pathogenic potential of premise plumbing-associated OPs specifically. Copper resistance is important to mammalian host infection for *P. aeruginosa* [311] and *A. baumannii* [312,313], and other evidence suggests that exposure to copper in aquatic environments selects for greater copper resistance among certain OPs [129,303,304]. Copper and other divalent metals may also play a role in nutrient acquisition and pathogenesis even after infecting hosts [314]. 

The effects of iron exposure on OPs are not as apparent. *L. pneumophila* serogroup 1 grown in medium that was iron limited (0.017–0.056 mg/L) has been shown to lose its virulence [315], indicating that limiting adequate concentrations of iron could not only decrease the presence of *Legionella* but also the likelihood of human infection. Iron also plays a role in modulating various behaviors, including modulating virulence factor production in *P. aeruginosa* and *A. baumannii* [316,317,318,319,320,321], but it is unclear what effects exposure to iron have on virulence in the premise plumbing environment. This subject is largely unexplored and more research is needed to determine the overall effects of the premise plumbing environment on OP virulence.

### 5.5. Antibiotic Resistance and Tolerance

Copper, among other heavy metals has been shown to exert selection pressure, leading to enhanced survival of antibiotic resistant bacteria. In fact, heavy-metal-associated co-selection and cross-selection has been proposed to be as much of a concern for environmental propagation of antibiotic resistance as antibiotics themselves [322]. Increases in antibiotic resistance genes at the community scale have been identified after long-term copper exposure in soil [323,324,325,326], sediment [327], and drinking water [327]. Bench-scale tests using bacterial isolates from biofilters [328] and wastewater [329] inoculated into growth media have shown that a selective or inductive effect of copper can take places within hours. However, these studies were performed with copper concentrations 5–77 times greater than the 1.3 mg/L US EPA copper action level and similarly in exceedance of the Chinese Standard for Drinking Water Quality of 1 mg/L [209] and WHO Guideline for Drinking-Water Quality of 2 mg/L [82]. Thus, these concentrations may not be representative of potable water systems. One study examining antibiotic resistant and sensitive strains of *Staphylococcus aureus* showed that the more antibiotic resistant strain survived longer in a copper container [90]. As discussed above, copper may also better support *Acanthamoeba* than other materials, while in one study *L. pneumophila* grown within *A. polyphaga* demonstrated increased tolerance to all antibiotics tested (rifampin, ciprofloxacin, and erythromycin) compared to those grown in culture media [330]. The role of copper plumbing and other pipe materials in these emerging areas of research is worthy of further investigation. 

There is more limited evidence that the presence of iron may also induce or select for antibiotic resistance, as observed for *P. aeruginosa* using iron-amended growth media [330] and the gut microbiomes of mice supplied with iron-amended water [331]. The latter case, while using an iron concentration more than 25 times the EU drinking water standard of 0.2 mg/L [332] and 16 times both the US EPA National Secondary Drinking Water Standard and Chinese Standard for Drinking Water Quality of 0.3 mg/L, may be of particular concern, as it suggests that pipe corrosion products have the potential to select for antibiotic resistance inside the infected host organism.

## 6. Conclusions

Premise plumbing is a complex, temporally dynamic, and spatially diverse environment that is strongly influenced by pipe materials. Virtually all pipe materials have known benefits and/or detriments for OP growth. Plumbing materials are an important driver of the chemical and biological water quality parameters that influence the control of OPs and there are no silver (copper or plastic) bullets that will uniformly inhibit the growth of *Legionella* and other OPs under all circumstances. 

Synthetic plastic pipe materials vary between type and manufacturer. They can act as a supply of organic carbon for the growth of microorganisms, but exert a lower chlorine demand and tend to form fewer scales that could provide more surface area for biofilm growth. Iron pipes supply nutrients for growth, exhibit a high disinfectant demand, produce hydrogen and other nutrients through corrosion, and tend to form thick scales with extremely high surface areas. While they may no longer be used in new construction, even short sections of pipe can affect an entire downstream premise plumbing distribution system. Stainless steel has few known effects on water quality, and correspondingly, OP control, perhaps because it is the least studied and is less commonly used as a result of its high cost. Copper pipes are known for their antimicrobial ability, but this is inconsistently realized in practice, and in some cases they seem to encourage OP growth relative to other pipes. Premise plumbing materials have a role to play in preventing OP infections and, at a minimum, should be examined more closely for their propensity to inhibit or stimulate OP proliferation during outbreak investigations. Research is needed to better define: Both the intra-species and inter-species variation of copper resistance amongst OPs, as well as environmental drivers of this variation.Effects of copper pipes on OPs in a more holistic sense, with identification of real-world conditions that are drivers for discrepancies in copper’s antimicrobial capacity.Copper’s possible micronutrient activity in OPs within premise plumbing contexts, including threshold concentrations required for various physiological functions, as well as physicochemical and ecological factors that influence those thresholds.The disease risk that VBNC OPs pose and conditions under which copper and other antimicrobials induce VBNC status in premise plumbing OPsThe inhibitory action of trace metals on OP growth in premise plumbing, as well as growth requirements for other trace elements exhibited by OPs in premise plumbing.Potential mediating effects of the wider microbial community composition resulting from pipe material on OPs.Effect of mixed pipe materials on physicochemical parameters of bulk water and OP growth.The effects of plumbing materials on OP antibiotic resistance and virulence.The impact of stagnation, velocity, sediments, corrosion control, and consumer water use patterns on all of the above.

An improved understanding will provide actionable advice for multiple stakeholders. In addition to the obvious direct use of the results in the construction industry and by building water quality managers, water utilities can benefit from improved understanding of how the interplay of premise plumbing pipe materials with disinfectants, nutrients and corrosion control can be harnessed to reduce disease incidence. 

## Figures and Tables

**Figure 1 pathogens-09-00957-f001:**
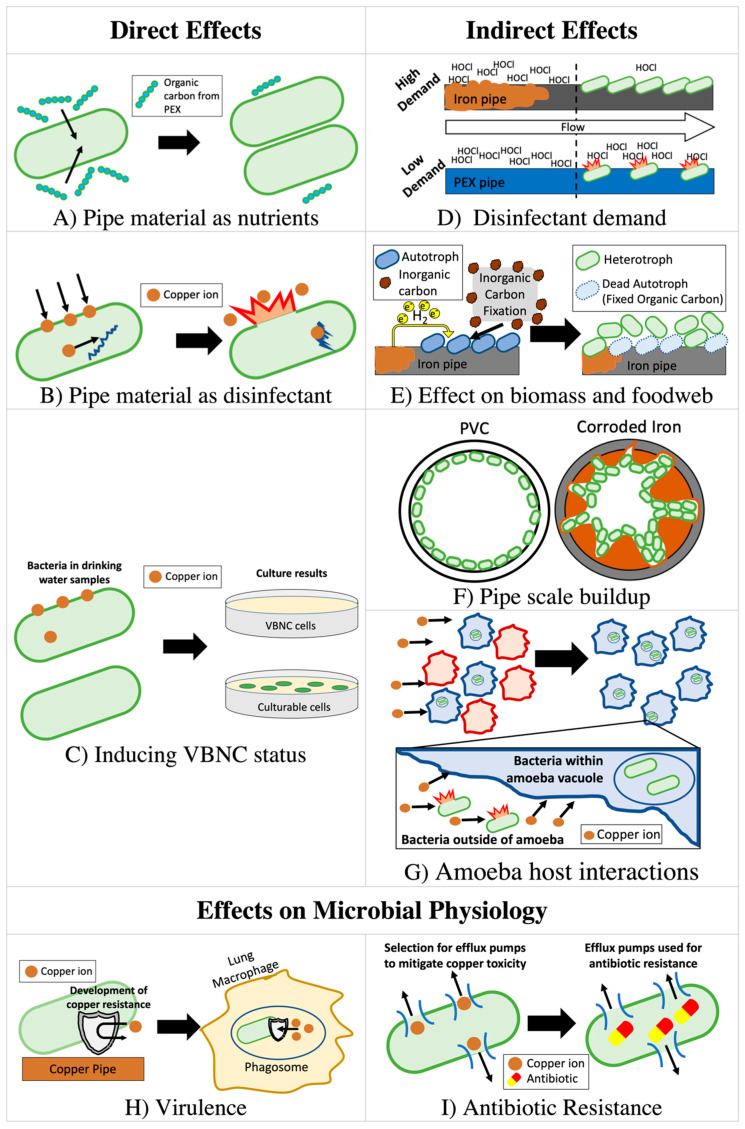
Overview of exemplar mechanisms by which pipe materials can affect OPs in premise plumbing. Depending on the circumstances, the pipe material itself can have direct effects on OPs growth by: (**A**) providing organic or inorganic nutrients that enhance growth, (**B**) acting as a growth-inhibiting antimicrobial, or (**C**) inducing viable-but-non-culturable (VBNC) status, from which microbes might recover in terms of infectivity and growth rates subsequent to exposure. Pipes can also indirectly affect OPs by: (**D**) consuming secondary disinfectants, allowing for microbial growth downstream, (**E**) evolving hydrogen gas or enhance nitrification, fueling autotrophic growth, or (**F**) developing thick pipe scales, which provide additional surface area for microbial growth, or (**G**) selecting for certain types of amoebae that are preferred hosts for bacterial OPs and protect them from negative effects of copper and disinfectants. Finally, pipes may unfavorably alter the physiology of microbes by increasing (**H**) OP virulence by selecting for resistance to phago-somal copper overload, or (**I**) resistance to antibiotics.

**Figure 2 pathogens-09-00957-f002:**
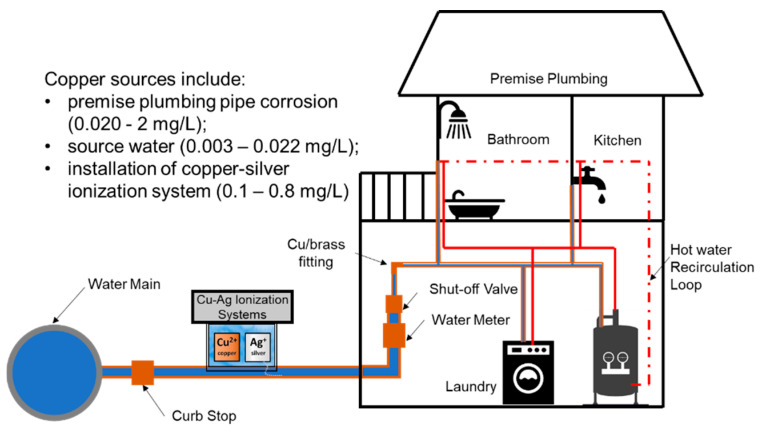
Copper sources in premise plumbing [81,82,83,84]. Note that Cu-Ag Ionization systems can be used in either point of entry or hot water distribution networks.

**Figure 3 pathogens-09-00957-f003:**
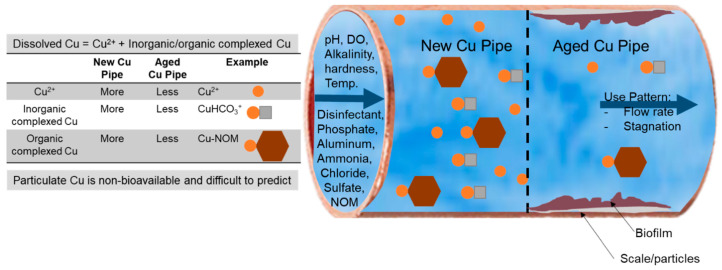
Copper pipe corrosion and speciation is controlled by influent water chemistry and pipe age. Water chemistry parameters, such as pH, dissolved oxygen (DO), disinfectants, inorganic complexing agents (e.g., alkalinity, phosphate, and ammonia), organic complexing agents (e.g., natural organic matter (NOM)), hardness, trivalent metal ions (e.g., aluminum, iron), sulfate, and chloride can influence copper pipe dissolution, speciation, and the precipitation process. Copper is categorized as either free copper ions and inorganic complexed copper (considered relatively bioavailable), or organically complexed or particulate copper (considered relatively non-bioavailable). The level of copper species in the premise plumbing systems are also affected by the pipe aging (new vs. old pipes) and the water use pattern, including flow rate, stagnation and temperature.

**Figure 4 pathogens-09-00957-f004:**
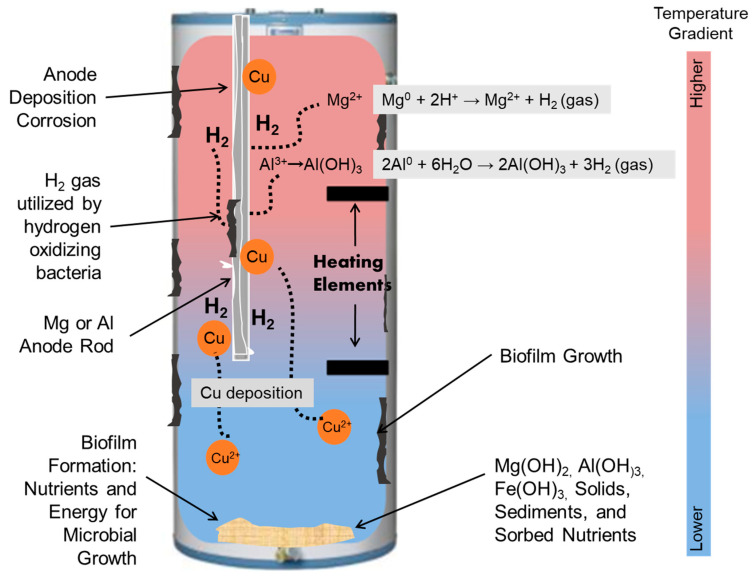
Water heater material interactions create multiple niches suitable for bacterial and opportunistic pathogen (OP) growth. Deposition of copper onto less noble metals (e.g., a water heater anode) can result in dramatically accelerated corrosion and release dissolved H_2_ gas, which is an electron donor for autotrophs. If the anode rod consists of magnesium, then the pH will become elevated as well. Figure adapted from Brazeau et al. [229].

**Table 1 pathogens-09-00957-t001:** Positive (+), Negative (-, --), and Neutral (0) Pipe Material Effects on OPs Control as Mediated by Various Water Chemistry Attributes.

Water Chemistry Attribute Influenced by Pipe Materials	Relevance to OPs	Effect of Pipe Materials on OPs Control as Mediated by the Indicated Water Chemistry Attribute				
		Copper	PVC	PEX	SS	Iron ^1^
Chlorine	Disinfectant	- [43]	0 [43,44,45,46,47,48,49,50]	- [43,51,52]	0/- [43,44,45,48,53]	-- [43,44,45,46,47,48]
Chloramine	Disinfectant	- [43,54]	0 [43,50]	0 [43,52]	0 [43]	-- [43,55,56]
Assimilable Organic Carbon	Carbon source	0	- [42,56,57]	-- [42,56,58,59]	0	0
Hydrogen Gas (aq)	Food web	0	0	0	0	- [60,61]
Release of Metals	Release of metals	+/- [59,62,63,64]	0	0	0 [65]	-- [66]

**Abbreviations:** OPs, opportunistic pathogens; PVC, Polyvinyl chloride; PEX, cross-linked polyethylene; SS, stainless steel; aq, aqueous. **^1^** Includes unlined iron and old galvanized iron pipes.

**Table 2 pathogens-09-00957-t002:** Copper can be growth-promoting or -inhibiting to opportunistic pathogens.

Opportunistic Pathogen	Associated Diseases	Exposure Route(s)	Inactivation via Copper		Growth via Copper	
			*Antimicrobial Efficacy **	*Evidence for Cu-Induced VBNC*	*Micronutrient Activity*	*Amoeba-Mediated Growth*
*Amoebae*	Encephalitis, Eye infections, Primary amebic meningoencephalitis [85,86,87]	Dermal, Inhalation, [85,86,87]	Moderate to Somewhat inhibited [59,88]	Unknown and unlikely	Possible that organisms are copper deficient and additional copper could increase growth [59,80]	NA
*Acinetobacter baumannii*	Bacteremia, Meningitis, Pneumonia, Urinary tract infections [89]	Dermal, Inhalation [89]	Moderate to Somewhat inhibited [58,90,91,92]	Unknown		Yes [93,94]
*Staphylococcus aureus*	Bacteremia, Endocarditis, Osteomyelitis, Pneumonia, Sepsis, Skin infections [95]	Dermal, Inhalation [96,97]	Moderate [90,98,99]	Unknown		Yes [100,101]
*Stenotrophomonas maltophilia*	Bacteremia, Endocarditis, Eye infections, Meningitis, Pneumonia, Sepsis, Skin infections, Urinary tract infections [102,103]	Dermal, Inhalation [102,103]	Moderate [91,92]	Limited [104]		Yes [105]
*Nontuberculous Mycobacteria (NTM): Mycobacterium avium complex; Mycobacterium abscessus complex; Mycobacterium kansasii and other species*	Bacteremia, Pneumonia, Skin infections [106]	Dermal, Ingestion, Inhalation [107]	Moderate [108,109,110,111]	Limited [109]		Yes [36,112]
*Aeromonas hydrophila*	Gastroenteritis, Meningitis, Peritonitis, Pneumonia, Skin infections [113]	Ingestion, Inhalation [113]	Unknown [114]	Unknown		Yes [115,116]
*Legionella pneumophila*	Legionnaires’ disease, Pontiac fever [117]	Inhalation [118]	Somewhat inhibited to High [62,83,119,120]	Moderate [121,122]		Yes [123,124]
*Pseudomonas aeruginosa*	Bacteremia, Endocarditis, Eye infections, Gastroenteritis, Osteomyelitis, Pneumonia, Sepsis, Skin infections, Urinary tract infections [125]	Dermal, Ingestion, Inhalation [125,126]	Somewhat inhibited to High [90,91,92,98,99,127,128,129,130,131,132]	Strong [127,132,133]		Yes [36,134]

* Categorizations of efficacy based upon studies that showed planktonic phase growth inhibition at: <0.1 mg/L (High), 0.1–0.8 mg/L (Moderate), and >0.8 mg/L (somewhat inhibited) copper concentrations in water or media.

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
