# Peer review of "Critical Review: Propensity of Premise Plumbing Pipe Materials to Enhance or Diminish Growth of Legionella and Other Opportunistic Pathogens"

_pathogens, 2020, doi:10.3390/pathogens9110957_

Round 1

Reviewer 1 Report

The paper is a review of how premise plumbing pipe materials might affect the presence of opportunistic pathogens. Of pipe materials, the review focuses on copper although also iron and plastic pipes are covered. Of pathogens, the main focus of the review is on Legionella.

With a substantial number of references cited (n=322), the review would benefit from adding a Table of contents. Overall, the writing style seems still a bit unorganized. For some reason it was difficult to follow the path of story when reading the review. Between the paragraphs some unnecessary repetition was present (detailed below) and I was not able to detect the main idea of the paragraphs from the starting sentences - sometimes the main point was presented at the end of the paragraph.

While the conditions in the United States are presented as examples in some occasions, the authors should consider also the readers from other parts of the world, as the presence of opportunistic pathogens within the premise plumbing microbiota is a worldwide public health concern. Furthermore, in some parts of the text, the link to the pathogen growth is missing and should be mentioned (especially in chapter 2).

Point-by-point notes:

Title: are the words "Critical Review:" necessary?

Table 1: The table is not well developed yet and the qualitative indications are difficult to follow. In the footnote, what is meant by "or as yet unknown (?).*", the question mark is not present in the table. Further "& Includes" remains unclear. It seems SS column should be before Iron column. The authors should consider using arrows up/down instead of plus/minus sign to increase clarity?

Figure 1: This is very broad mixture of everything. Seems to have a focus in copper and therefore the figure legend mentioning PEX as and example. It is not clear if the orange dots in the figures 1c, 1e, 1g, 1h and 1i all mean copper like in figure 1b.

Page 5: are the individual sentences meant to be as a bulleted list? It does not read well as a text paragraph. Of the topic presented, I propose to add lack of binding standardization for plastic products. Further, conflicting UN sustainable development goals could be mentioned too.

1.1: Copper as an Antimicrobial Agent?

Table 2. Check where bold text is used and fill the micronutrient activity column which seems mainly empty. Explain UTI? In the footnote, it would be good to mention if mg/L denotes for copper concentration in water.

1.2 Copper Pipe as an Antimicrobial Material?

Throughout: check the use of CAPITAL letters, see for example titles of Chapters 2,3 and 4.

Chapter 3. The preface to this chapter seems to be in an incorrect place as it refers to the "described above" (page 13).

3.2: control measures for controlling (not disinfecting).

3.4: at the end of paragraph there is unnecessary repetition to the earlier parts - please check!

4.1: the mentioned "some doubt" would require adding a reference.

Page 15: on the fifth line, there is one study mentioned, but with four references - please check!

Page 16: Eleven references in one sentence seems too many - maybe the main ones kept as an example?

Chapter 5: It seems that there should be a bullet point list, but the bullets are missing?

References: This section requires editing, check especially references numbered as 64, 120, 168, 169, 187, 190, 257, 260, 264, 272 and 308. 

Author Response

Thank you for your time and effort in evaluating our manuscript and offering suggestions for improvement. Please see the attached response.

Reviewer 2 Report

In this review the AA performed a specific and accurate observation regarding realtionship between pipes matherials and waterborne pathogens growth.

This work is complete and argued with a lot of references.

I would suggest some integrations.

1) point 1.5. Please, it is important define the correlations between zinc presence and Nontuberculous mycobacteria.

2) point 3.1 You may describe some strategies aimed to redice water stagnation (automized water flushing).

Author Response

(The authors gave the same response as above.)

Round 2

Reviewer 1 Report

The authors have made good work in revising the manuscript, it reads now much better - the story is more clear. In my opinion, some of the improvements, however, still require minor editing as detailed below.

For me, Table 2 remains still unclear. In particular, it is difficult to follow what is the negative effect on disinfectant decay, carbon source, food web or release of metals. Does the decay become smaller, is there less carbon or released metals? There might be too many variables here - it seems that the affecting parameters are thought to reduce OPs levels, but that is not clear nor written out in the table header referring to "different water quality parameters". Further instead footnote "2", maybe better to use: "Abbreviations: PVC, polyvinyl chloride, PEX, cross-linked poyethylene and SS, stainless steel." 

In Figure 1, there seems to be three groups of illustrations: Direct effects, Indirect effects and Effects on Microbial Physiology. Please include these groups to the text of the figure legend for clarity: ..pipes can have direct effects on OPs by A) providing..., B) acting..., and C) inducing ... . Pipes can also have indirect effects: D, E, F, and G. Furthermore, the pipes may affect OP virulence by H or I. 

Figure 2. Please note that Cu-Ag ionization systems might operate in some buildings only in hot water tubes. It is also noteworthy that European Union does not accept copper disinfection and special approval for such systems are needed prior installation and operation (see: https://pubmed.ncbi.nlm.nih.gov/23763088/).

Table 2. Check why Encephalitis... is in bold? In the footnote, add: "...(Somewhat inhibited) copper concentrations in water or media."

Page 8, rows 200-201 and elsewhere: the numbers 23950, 15063 should be rounded, usually two meaningful numbers are enough, i.e. 24000, 15000. Further, the range of Legionalla from < 10 to 110 CFU/L is better than presenting negative numbers (-90 CFU/L does not seem right).

Page 8, Row 209: strain (not stain)?

Page 9, Row 253: specify what is meant by CDC biofilm reactors?

Page 11, row 320 and elsewhere: please specify EPA - is United States Environmental Protection Agency meant here? The authors state in their response that they wish to write their review to the international audience. However, the legislation examples (all except one) seem to refer to the US national regulations.

Page 13, Row 391: ...sections are further...?

Page 13, Row 422: dead ends (instead of dead-legs?)

Page 14, Row 454: italize Legionella

Page 17, Row 595: other heavy metals

Page 18, Row 637: the line "Research is needed to better define:" does not seems to be part of the first bullet point, but should be on its own row - please check!   
